# A Study on Webtoon Generation Using CLIP and Diffusion Models †

**Kyungho Yu [1], Hyoungju Kim [2], Jeongin Kim [3], Chanjun Chun [1] and Pankoo Kim [1,*]**

[1] Department of Computer Engineering, Chosun University, 309 Pilmun-Daero, Dong-Gu, Gwangju 61452, Republic of Korea; infinitegh@chosun.ac.kr (K.Y.); cjchun@chosun.ac.kr (C.C.)
[2] Institute of AI Convergence, Chosun University, 309 Pilmun-Daero, Dong-Gu, Gwangju 61452, Republic of Korea; hyoungjukim@chosun.ac.kr
[3] Department of Microbiology and Immunology, Chosun University School of Dentistry, 309 Pilmun-Daero, Dong-Gu, Gwangju 61452, Republic of Korea; jungingim@gmail.com
* Correspondence: pkkim@chosun.ac.kr
† This paper was written based on the foundation of Kyungho Yu's Ph.D. Thesis.

**Abstract:** This study focuses on harnessing deep-learning-based text-to-image transformation techniques to help webtoon creators' creative outputs. We converted publicly available datasets (e.g., MSCOCO) into a multimodal webtoon dataset using CartoonGAN. First, the dataset was leveraged for training contrastive language image pre-training (CLIP), a model composed of multi-lingual BERT and a Vision Transformer that learnt to associate text with images. Second, a pre-trained diffusion model was employed to generate webtoons through text and text-similar image input. The webtoon dataset comprised treatments (i.e., textual descriptions) paired with their corresponding webtoon illustrations. CLIP (operating through contrastive learning) extracted features from different data modalities and aligned similar data more closely within the same feature space while pushing dissimilar data apart. This model learnt the relationships between various modalities in multimodal data. To generate webtoons using the diffusion model, the process involved providing the CLIP features of the desired webtoon's text with those of the most text-similar image to a pre-trained diffusion model. Experiments were conducted using both single- and continuous-text inputs to generate webtoons. In the experiments, both single-text and continuous-text inputs were used to generate webtoons, and the results showed an inception score of 7.14 when using continuous-text inputs. The text-to-image technology developed here could streamline the webtoon creation process for artists by enabling the efficient generation of webtoons based on the provided text. However, the current inability to generate webtoons from multiple sentences or images while maintaining a consistent artistic style was noted. Therefore, further research is imperative to develop a text-to-image model capable of handling multi-sentence and -lingual input while ensuring coherence in the artistic style across the generated webtoon images.

**Keywords:** text-to-image; multimodal AI; webtoon; diffusion model

## 1. Introduction

Advancements in deep learning (DL) technology have increased momentum in research on image generation through computer learning and autonomous image generation [1]. Image generation techniques have evolved from generative adversarial networks in which a generator and discriminator engage in adversarial training to produce realistic fake images that closely resemble the originals. At present, the use of diffusion models in image generation has led to the creation of remarkably realistic images that are indistinguishable from those drawn by humans. Recent developments in DL-based image generation have found applications in the entertainment sector, such as virtual character creation and animation, owing to their ability to generate images resembling those handcrafted by humans [2,3].

Concurrently, advances in text-to-image generation technology have enabled computers to replicate the process of associating images with textual descriptions [1,2]. DL-based text-to-image generation involves feeding textual descriptions into a DL model to generate corresponding images. This is feasible because the process of training image generation models allows them to learn the relationship between text and images, thereby facilitating image synthesis from textual inputs. Text-to-image generation involves co-training both image and text data, known as "multimodal learning". This enables the model to generate output in a multimodal format by exploiting the learned relationships between diverse modalities. This emerging paradigm is referred to as multimodal artificial intelligence (AI) [4,5]. For instance, information about a dog can be represented multimodally by incorporating images of the dog, its barking sound, and its name. By training numerous multimodal instances of dogs, the model can generate images based on textual descriptions, or classify a given image as a specific type of dog, thereby exemplifying multimodal AI. OpenAI CLIP (contrastive language image pre-training) is an example of a technology that leverages contrastive learning to map text and image features into a shared latent space, thereby capturing the relationships between disparate domains.

Webtoons, a portmanteau of "web" and "cartoon", refers to online serialized comics presented in a vertical scroll format for ease of reading on mobile devices. Unlike traditional comics, webtoons have the unique feature of vertical scrolling layouts. The webtoon industry, led by Korean portal companies, has flourished domestically and internationally, yielding substantial profits and establishing itself as a high-value-added sector. Webtoon production involves various stages, including story conception, treatment creation, content production, sketching, inking, coloring, and background drawing [6]. Owing to the labor-intensive nature of each stage, recent advancements in AI technologies have been introduced to assist in webtoon creation. For example, Naver, a webtoon platform, introduced an "AI Painter" [7] that automates the sketching and inking process when the artist provides a basic outline. Moreover, the research has explored the generation of webtoons from textual descriptions using Text-to-Image models [8–10]. Through analyzing the adoption process of innovative technologies in webtoon creation based on factors such as relative advantages, suitability, complexity, trialability, and observability, the innovation diffusion theory highlights the positive impact of AI-driven webtoon creation [6]. While AI technology can directly assist in creating webtoons, webtoon artists with a lack of creativity may find it difficult to visualize their ideas. Therefore, research has been conducted on the generation of webtoons from textual descriptions, known as treatments, using text-to-image models. However, previous studies primarily relied on GAN-based approaches to create webtoons from text, resulting in low-quality images and limitations in the generation of diverse webtoons [10]. Consequently, there is a need for technology that can generate high-quality webtoons from text inputs, providing valuable assistance to artists in realizing their webtoon concepts.

This study endeavors to advance the field of webtoon creation by employing DL-based text-to-image techniques and using CLIP and diffusion models to generate webtoons from textual descriptions. A treatment webtoon dataset was constructed and trained using CLIP to learn the relationships between the treatments and webtoon images. This process involves inputting textual descriptions into CLIP to locate the most similar images within the dataset. Subsequently, to generate webtoons, treatments, along with images identified by CLIP, were placed into a pre-trained diffusion model.

The rest of this paper is outlined as follows. Section 2 introduces relevant studies related to this research. Section 3 outlines the training of the multimodal treatment webtoon dataset on CLIP, facilitating cross-domain relationships and guided image generation using the identified images. Section 4 describes the proposed approach, using CLIP and the diffusion model for webtoon generation. Finally, Section 5 concludes the study, highlights its limitations, and suggests directions for future research.

## 2. Related Work

### 2.1. Diffusion Models

The diffusion model is a probabilistic modeling approach used to generate images. This model encompasses the modeling of an image's probability distribution, enabling new images to be created using this framework [11]. The diffusion model leverages a probabilistic approach that has the advantage of yielding diverse outcomes during image generation. In addition, owing to its ability to reuse learned information during the image-generation process, the diffusion model enhances the efficiency of image synthesis.

The diffusion process employed in denoising diffusion probabilistic models comprises two key components (see Figure 1). One is the forward-diffusion process and the other is the reverse-diffusion process. The forward-diffusion process depicted from right to left in Figure 1 involves the gradual introduction of noise into the pixel values of an image at each step, thereby diffusing the image. In contrast, the reverse-diffusion process, depicted as moving from left to right, aims to reverse the diffusion and restore the original image from the noisy version. Repeating these two processes culminates in the creation of new images from the noise.

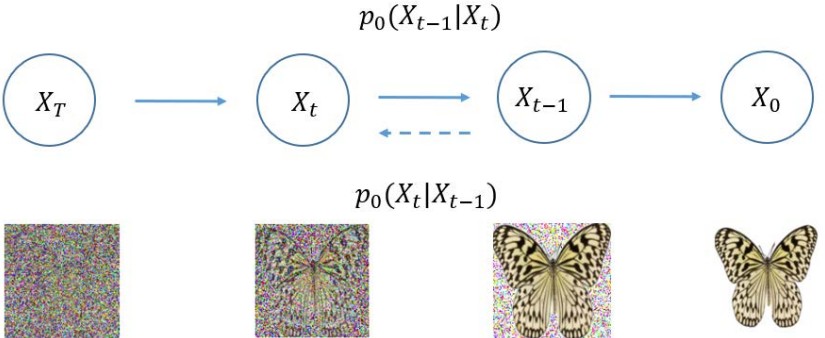

**Figure 1.** The training process of the diffusion model.

Training the diffusion model involves the use of extensive image datasets. During this process, the model learns by minimizing the discrepancy between the images generated through the reverse diffusion process and actual images. Although the forward-diffusion process was not used during training, it was employed in the image-generation phase to introduce noise.

Text-to-image generation models based on the diffusion model include notable examples such as DALL-E 2 and the stable diffusion model [12,13]. In addition, DALL-E 2 is known as unCLIP. Comprising CLIP, a prior component, and a decoder, DALL-E 2 learns from extensive text–image pairs in large web datasets, leveraging the interaction (similarity) between the text and images based on CLIP training. To generate images, DALL-E 2 employs pre-trained CLIP, prior component, and decoder. When generating an image, the text describing the desired image is input into the text encoder of the CLIP model to create CLIP text embeddings. The first stage employs a diffusion-model-based prior, inputting CLIP text embeddings into the diffusion prior to generate CLIP image embeddings. The decoder adopts a modified guided language-to-image diffusion for generation and editing [14]. Within DALL-E 2, the decoder uses four inputs, encoded text, CLIP text embeddings, noisy CLIP image embeddings, and timestep embeddings, which generate images.

The stable diffusion model, which is similar to the diffusion model, consists of a CLIP-based text encoder, an information generator consisting of UNet, a scheduler, and a decoder for image generation. Note that the stable diffusion model generated images in the latent space rather than in the pixel space, leading to rapid image generation. Recent advancements in DL-based generative models extended beyond text-to-image creation, with diffusion-model-based approaches finding applications across diverse domains [15–18]. Table 1 is a comparison table of the performance of Dalle-2, Stable Diffusion, and GLIDE.

**Table 1.** Comparison of FID scores for three DIFFUSION models. The Frechet Inception Distance (FID) scores measure the difference between the distributions of real and generated images. (Lower scores indicate better performance.)

| Model | Company | FID |
|---|---|---|
| Dalle-2(unclip) [12] | OpenAI | 10.39 |
| Stable Diffusion [13] | Stability AI | 8.59 |
| GLIDE [14] | OpenAI | 12.24 |

*2.2. Multimodal Learning*

Multimodal learning refers to the representation of a single piece of information using multiple types of data, encompassing various modalities such as images, text, and audio, to extract and learn features. The CLIP model introduced by OpenAI is an example of a multimodal learning model [19]. CLIP learns features from both images and text simultaneously. It positions the features of images and text within the same latent space of dimensions and employs contrastive learning to analyze the relationships between these features. Contrastive learning involves comparing data to position similar data features closer together and dissimilar features further apart. This enabled an understanding of the relationships between data from different domains.

One notable advantage of CLIP is its ability to place multimodal data in a shared latent space, thereby facilitating the learning of relationships among different modalities. CLIP establishes connections between images and text, allowing for similarity measurements. Building on these strengths, models such as DALL-E 2 by OpenAI and Dreambooth by Google use CLIP to measure text–image similarity or as an encoder for extracting features from text or images.

The CLIP structure includes separate encoders to extract features from images and text, each of which is augmented with projection layers of the same dimensions. When text is input into CLIP, it undergoes text encoding to extract features and is embedded after passing through the final projection layer. Similarly, images are processed using the image encoder to extract features, followed by embedding after the projection layer using the same approach as in the text. CLIP training employs contrastive learning, extracts features with the same dimensions from each encoder, and subsequently normalizes them to calculate the cosine similarity. Then, the inner product of the text- and image-embedding vectors is computed. In the provided code, "logits_per text" constitutes a matrix of similarity values between text and images, while "logits per image" forms a matrix of similarity values between images and text. Subsequently, cross-entropy is computed to calculate the loss and update the respective networks during training.

Following training, the embedding vectors (feature vectors) for text and images could be computed independently. For instance, to obtain the CLIP embedding vector for the sentence "A man is walking down the street", one could input it into CLIP's text encoder to compute the embedding value. Images could be processed through the CLIP image encoder using a similar approach.

**3. Methods of Generating Webtoons Using CLIP and Diffusion Models**

*3.1. Multimodal Learning of Webtoon Datasets Using CLIP*

In this section, a multimodal learning approach was employed to measure the similarity between the data from distinct domains using the treatment webtoon dataset [10]. The CLIP model leveraged a transformer-based multilingual BERT and Vision Transformer (ViT) as encoders to extract features from text and images, respectively. A common 512-dimensional, fully connected projection layer was appended to the final layer of each encoder structure.

Figure 2 illustrates the method to obtain logits for a batch size of five. Here, logits represent the cosine similarity between the text and images. As Figure 3 shows, measuring the cosine similarity between text and images involved normalizing the feature vectors of

both modalities and computing the matrix product. Cosine similarity ranged from $-1$ to 1, with higher values indicating greater similarity.

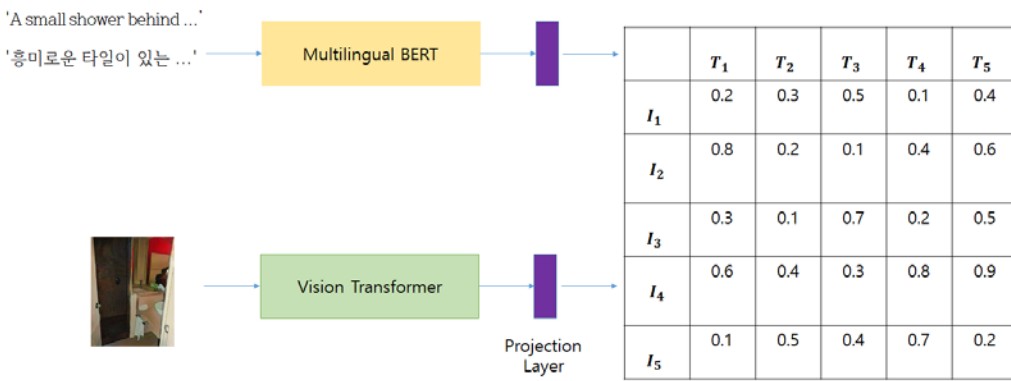

**Figure 2.** The structure of CLIP and an example of the logits matrix for a text–image pair.

```
# normalized features
image_embeds = image_embeds / np.linalg.norm(image_embeds, axis=-1)
text_embeds = text_embeds / np.linalg.norm(text_embeds, axis=-1)

# cosine similarity as logits
logits_per_text = matmul(text_embeds, image_embeds.T) * logit_scale
logits_per_image = logits_per_text.T
```

$$similarity = \cos(\theta) = \frac{A \cdot B}{\|A\|\|B\|}$$

**Figure 3.** Calculation method of consine similarity in the pseudocode of CLIP.

The accuracy measurement involved calculating the logits between the features of the text and images, followed by a comparison with labels. On the right side of Figure 3 the pseudocode outlines the accuracy computation using the previously computed logits. The correct label could be derived using the numpy "arrange" function. For instance, if there were five logits in Figure 3, executing the np. arrangement would yield a matrix [0, 1, 2, 3, 4]. This matrix signified the values at indices 0, 1, 2, 3, and 4 in each row. The p. argmax function identified the index with the highest value in each row, with the second argument being one for rows and zero for columns. In acc_i, the result of np.argmax(logits, 1) was an array such as [2, 0, 2, 3, 3], indicating that the largest values were found at indices 2, 0, 2, 3, and 3 in each row. Comparing the label array [0, 1, 2, 3, 4] with the argmax result array [2, 0, 2, 3, 3], the "==" operator yielded [False, False, True, True, False], resulting in an accuracy of 0.4, where the prediction matched the correct answer.

The choice of a label array [0, 1, 2, 3, 4] in Figure 4 stemmed from the fact that logits (blue portion in the left table of Figure 4) with the same indices explicitly represented data similarities during training, necessitating cosine similarities approaching one. Therefore, the logits indexed by the label array had to reflect the highest values in each row. Accuracy signified the proportion of agreement between the learned logits and correct labels throughout the training.

To calculate the CLIP loss, computed logits were used to calculate cross-entropy in both the row and column directions, followed by averaging, resulting in CLIP loss. Finally, the computed loss is used to update each encoder for text and image, enabling them to learn the features of similar data to be closer and the features of dissimilar data to be farther apart within the CLIP latent space.

| | $T_1$ | $T_2$ | $T_3$ | $T_4$ | $T_5$ |
|---|---|---|---|---|---|
| $I_1$ | 0.2 | 0.3 | 0.5 | 0.1 | 0.4 |
| $I_2$ | 0.8 | 0.2 | 0.1 | 0.4 | 0.6 |
| $I_3$ | 0.3 | 0.1 | 0.7 | 0.2 | 0.5 |
| $I_4$ | 0.6 | 0.4 | 0.3 | 0.8 | 0.9 |
| $I_5$ | 0.1 | 0.5 | 0.4 | 0.7 | 0.2 |

```
labels = np.arange(len(logits))
acc_i = ((np.argmax(logits, 1) == labels)).mean()
acc_t = ((np.argmax(logits, 0) == labels)).mean()
acc = (acc_i + acc_t) / 2
```

**Figure 4.** Method for calculating accuracy in the pseudocode of CLIP.

### 3.2. Webtoon Generation Using Diffusion Models

To generate webtoons using the diffusion model, the CLIP model trained in the previous section and a depth-to-image model of stable diffusion were employed [20]. The webtoon generation process involved two main steps. First, the image most similar to the desired treatment was identified within the dataset. Second, the most similar image was input, along with text, into the depth-to-image model to generate a webtoon.

Feeding text into the CLIP model to find similar images involved computing the cosine similarity between the CLIP embedding vectors of all dataset images and the embedding vector of the given text. The index of the embedding vector with the highest similarity score corresponded to the most similar image. Calculating the CLIP embedding values for all images was time-consuming. Therefore, to expedite the similarity calculation, the precomputed embedding vectors for all images were stored. When calculating the similarity for the treatment's CLIP embedding vector, the stored embedding values of the entire image dataset were loaded to measure similarity.

After identifying the text's most similar image within the dataset, a depth-to-image model was employed to generate the image. The depth-to-image model used the depth information of the input image as an initial image and progressively removed noise through an inverse diffusion process to create an RGB image. This approach was viable because the model was trained using paired RGB and depth images. During training, the model predicted the color information associated with the depth information and used it to determine the RGB values of the individual pixels. Here, because the depth-to-image model excelled in producing realistic images, the desired cartoon-style images were generated by inputting the model with the keyword "webtoon" as a query.

## 4. Experiment

In this section, we trained the CLIP model on the treatment webtoon dataset and used the diffusion model to generate webtoons to validate the outcomes. The CLIP model leveraged pretrained multilingual BERT and ViT encoders for text and images, respectively, with an added projection layer of 512 dimensions in the final output of each encoder. During training, we optimized all layers of the encoders without freezing any layers. The contrastive loss function was employed by placing the text and image vectors in the same-dimensional projection layer, measuring the cosine similarity between the vectors, and computing the loss through cross-entropy. Notably, the key hyperparameters used in the training included 25 epochs, a learning rate of 0.00001, and a batch size of 128, distributed across four GPUs with a batch size of 32 per GPU. The data used for training consisted of data from MSCOCO, which were transformed into cartoon-style images using Cartoon-GAN. After excluding incomplete data, a total of 820,752 text–webtoon data pairs were utilized for training, with 405,040 text–webtoon pairs used as validation data.

To evaluate the CLIP model's performance during the training, we assessed its accuracy and loss. Figure 5 shows the accuracy and loss measured while training the treatment webtoon dataset on the CLIP model. Over 25 epochs (see Figure 5a), the accuracy of the training dataset reached 88%, and the evaluation dataset demonstrated an accuracy of 60% (see Figure 5c). Increasing the number of training epochs revealed an increasing trend in accuracy and a decreasing trend in loss, indicating an improved performance with more epochs. The CLIP model used in the study was trained for 25 epochs.

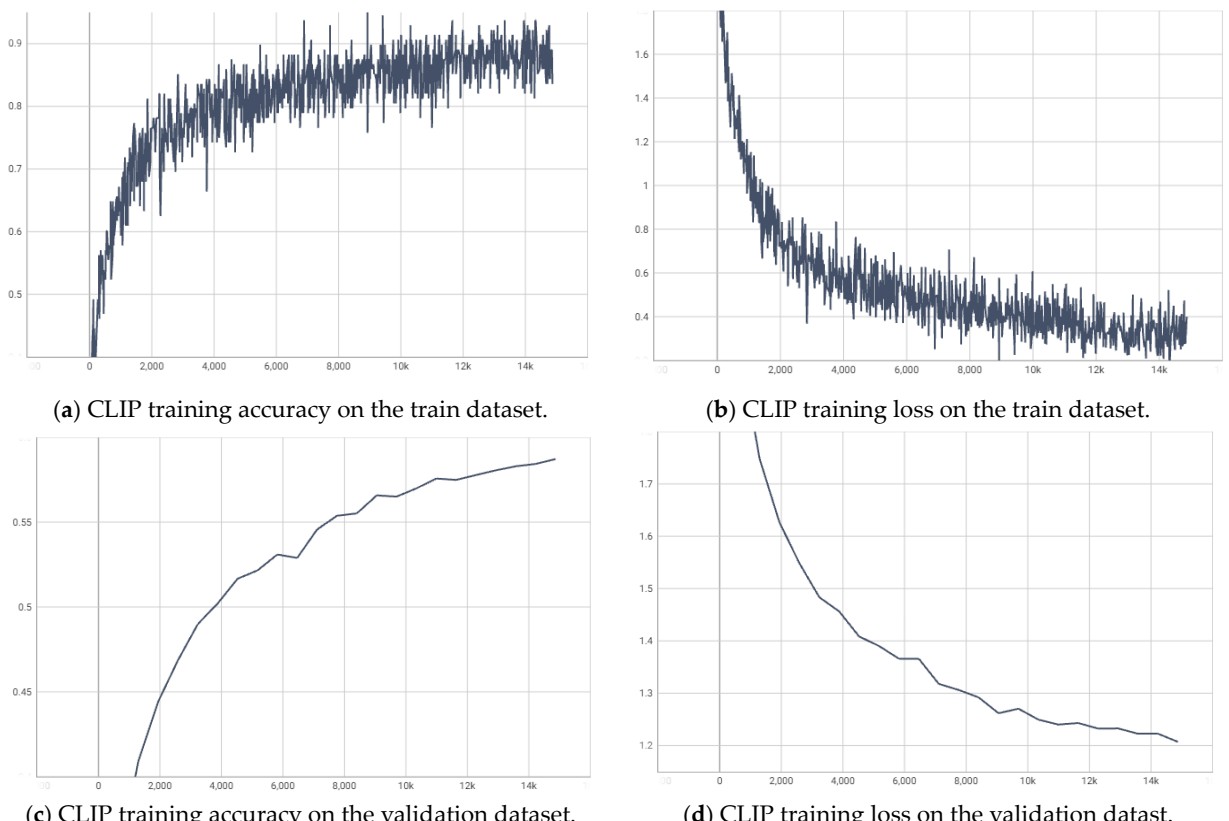

(**a**) CLIP training accuracy on the train dataset.

(**b**) CLIP training loss on the train dataset.

(**c**) CLIP training accuracy on the validation dataset.

(**d**) CLIP training loss on the validation datast.

**Figure 5.** Graph of accuracy (**a**) and loss (**b**) during CLIP training on the train dataset. Graph of accuracy (**c**) and loss (**d**) during CLIP training on the validation dataset. The x-axis of accuracy (**a**,**c**) represents the training steps, and the y-axis represents accuracy. For loss (**b**,**d**) the x-axis represents training steps, and the y-axis represents loss.

Subsequently, when text was used as a query, the dataset was searched for the most similar images. This process involved encoding the text using the CLIP model to obtain the text-embedding value, followed by measuring the cosine similarity between the precomputed CLIP-embedding values of the dataset images and the text-embedding value. Then, the top $N$ images with the highest similarity were output. Precomputing and storing CLIP-embedding values for dataset images were essential to expedite real-time similarity calculations, given the time-consuming nature of computing the embedding values for a large image dataset.

Table 2 presents the results of image retrieval when text was used as a query in the constructed dataset. Rows 1~2 of Table 2 display the outcomes when English treatments were used as queries, whereas rows 3~4 of Table 2 show the results for the Korean treatments. Image retrieval experiments revealed the ability to semantically find images related to the query and images resembling the original query.

**Table 2.** Top-3 most similar images when English and Korean texts are input to CLIP.

| Input Text | Ground Truth | Image Retrieval-1 | Image Retrieval-2 | Image Retrieval-3 |
|---|---|---|---|---|
| A horse-drawn carriage is parked along the curb. | 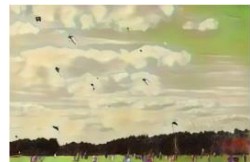 | 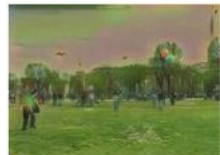 | 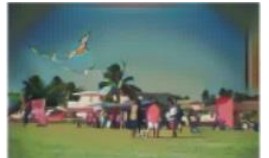 | 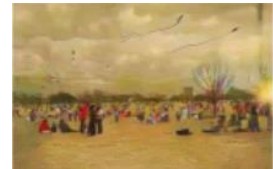 |
| A black bear sitting on rock with mossy patches. | 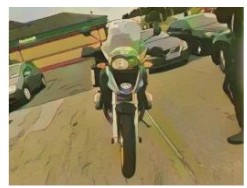 | 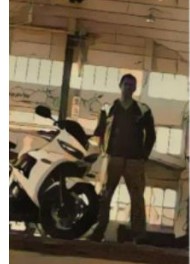 | 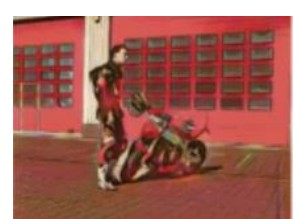 | 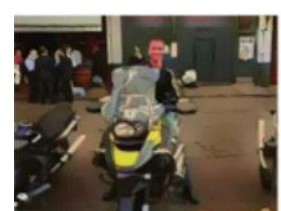 |
| 많은 사람들이 멋진 날을 즐기고 있어요, 큰 잔디밭에서 연을 날리면서 말이죠. (Many people are enjoying a wonderful day, releasing balloons into the air on a large grassy field.) | | | | |
| 주차장의 오토바이를 타고 옆에 서 있는 사람이 있다. (There's a person standing next to the motorcycle in the parking lot.) | | | | |

For webtoon generation using the diffusion model, a depth-to-image model of stable diffusion pretrained on the diffusion model was employed. Inputting both text and images into the depth-to-image model was essential for retaining the shape and guiding the generation process. By employing the depth information of the input images as the initial images and progressing through a reverse diffusion process to remove noise, the depth-to-image model created RGB images. Notably, cartoon-style images were generated by inputting the model with the keyword "webtoon" as the query. Table 3 shows the results of image generation using CLIP and the diffusion model:

(a) shows the input text in CLIP;

(b) shows the most similar image found in the webtoon dataset using CLIP;

(c) and (d) are the images generated by inputting the most similar image from CLIP into the depth-to-image model.

The generated images maintained the contextual features expressed in the input text and resembled the shapes of the most similar images.

Webtoons consist of vertically aligned continuous images. To generate semantically connected webtoons akin to actual webtoons, we used the constructed treatment webtoon dataset to generate sequential text data. By employing GPT-3.5-based ChatGPT [21], we generated sequences of three sentences each to represent successive text data. This method helped generate 5000 instances of sequential text data. The experiments were conducted in a manner similar to that of the previous approach. A single text was input to the CLIP

to find the most similar image, and the diffusion model generated webtoons based on the input text.

**Table 3.** Results of webtoon generation using CLIP and diffusion model.

| (a) CLIP Input Text | (b) Most Similar Image | (c) Generated Image 1 | (d) Generated Image 2 |
|---|---|---|---|
| Six teens playing frisbee in a field of grass | 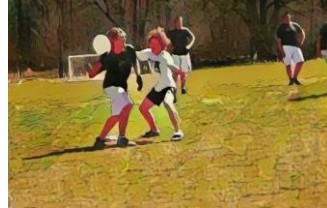 | 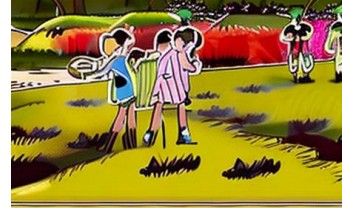 | 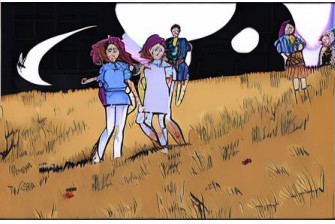 |
| A horse drawn carriage is parked along the curb. | 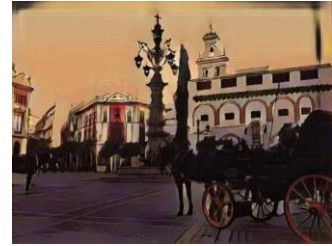 | 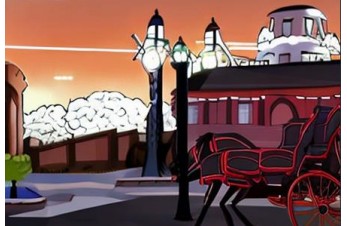 | 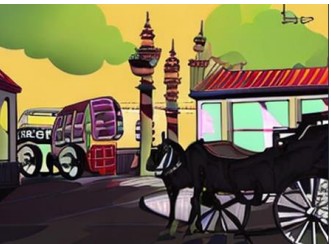 |
| Electric appliances crammed on a counter in a kitchen. | 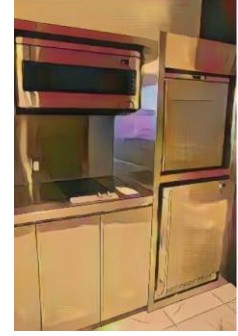 | 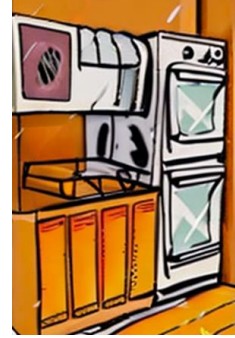 | 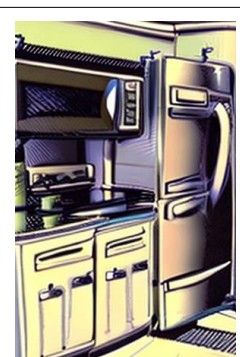 |
| A person in a ski suit is skiing. | 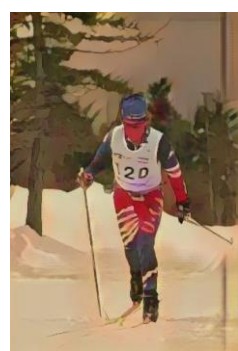 | 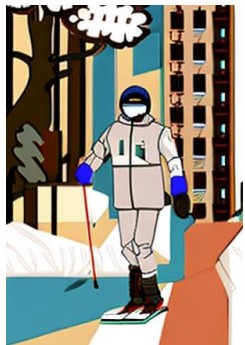 | 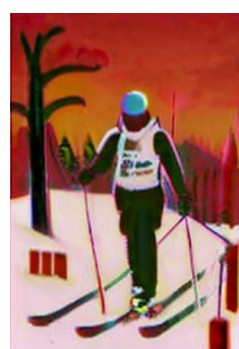 |

Table 4 displays the results of webtoon generation using sequential text data that were input into the CLIP and then the diffusion model. We generated webtoons by inputting all 5000 instances of sequential text data into CLIP and the diffusion model. The generated webtoons exhibited scene transitions similar to those of actual webtoons. For a quantitative evaluation, we measured the inception score for webtoons generated from sequential text data. Table 5 presents the measurement results. The Inception Score is a metric used to evaluate the image quality of generative models. It involves feeding the generated images into a pre-trained Inception model and calculating the probabilities of the images belonging to specific classes [22]. A higher Inception Score indicates better image quality,

and the experimental results reveal that the Inception Score for webtoons generated using CLIP and the Diffusion model from continuous text descriptions is 7.14.

**Table 4.** Results of webtoon generation using continuous text with CLIP and the diffusion model.

| (a) CLIP Input Text | (b) Most Similar Image | (c) Generated Image |
|---|---|---|
| Man riding a motor bike on a dirt road on the countryside. | 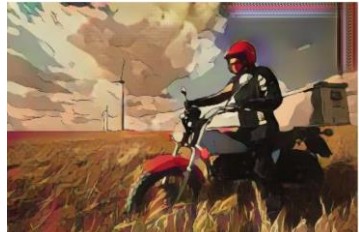 | 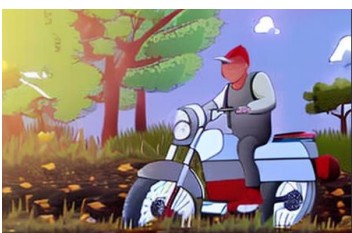 |
| The wind whipped through the rider's hair as he navigated the twists and turns of the path. | 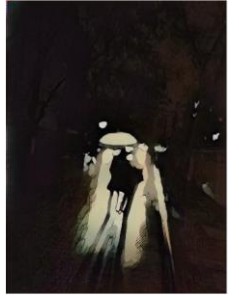 | 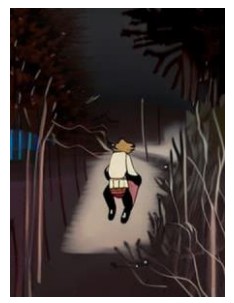 |
| The sound of the engine echoed across the fields, disturbing the peaceful silence of the countryside. | 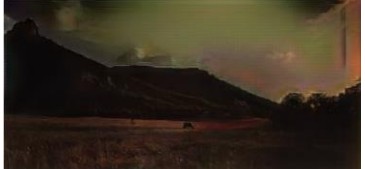 | 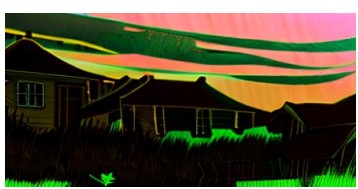 |
| He couldn't help but feel a sense of freedom as he sped along on his trusty machine. | 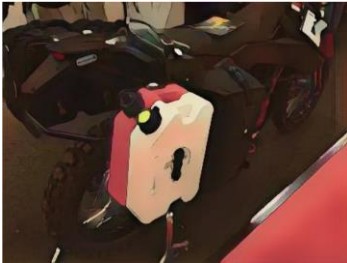 | 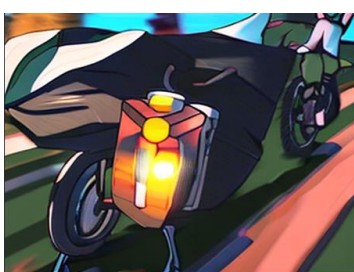 |

**Table 5.** Performance evaluation of the proposed text-to-image model.

| Datasets | Inception Score |
|---|---|
| Continues text data: 5000 | 7.14 |

## 5. Conclusions

In this study, we conducted research on the generation of webtoons using a DL-based text-to-image model to facilitate webtoon creation activities. To conduct our experiments, we constructed a treatment-webtoon dataset using the publicly available MSCOCO dataset and employed ChatGPT to generate continuous text data. To capture the relationship between multimodal data, such as the treatment webtoon dataset, we trained the CLIP model. CLIP extracted features from diverse domains and employed contrastive learning to ensure that features from same-dimensional data were similar, whereas those from different

data were distant. During training, one of the quantitative metrics, accuracy, and loss for both training and evaluation data increased as the number of training epochs progressed. However, after 25 epochs, the metrics began to decrease.

In the image retrieval experiment where the treatment was used as a query, we were able to identify images that reflected the features depicted in the actual images. To generate webtoons using the diffusion model, we input the text, CLIP features of the text, and similar CLIP image features into a pretrained depth-to-image model, which was based on the diffusion model. This approach led to the creation of webtoon images. We conducted two experiments for webtoon generation using the diffusion model: one involved generating webtoons using a single text, and the other involved generating webtoons using sequential text data. The results of webtoon generation using sequential text data revealed an inception score of 7.14. Moreover, the context and shape reflected in the guided text and images influenced the generated webtoons. These results could be attributed to the depth-to-image model, which used depth information to generate images. Furthermore, the generation process yielded diverse artistic styles with each iteration, which could be attributed to the depth-to-image model being pretrained on the LAION-5B dataset.

The multimodal webtoon generation conducted in this study, if employed by webtoon artists during their creative process, has the potential to expedite webtoon creation by enabling the generation of webtoons by simply inputting the desired text into a DL model.

However, some limitations of this study include the inability to generate webtoon-style images that consider multiple sentences and images, as well as the inability to ensure a consistent artistic style in the generated images. In addition, the CLIP model trained in this study could only handle both English and Korean inputs using multilingual BERT for tasks such as image retrieval and zero-shot classification. However, the pretrained diffusion model could only accept English inputs. In future work, we intend to explore models that can accommodate multiple sentences, maintain a consistent artistic style, and accept inputs in multiple languages.

**Author Contributions:** Conceptualization, C.C.; Methodology, H.K.; Software, K.Y. and J.K.; Validation, J.K.; Investigation, K.Y.; Resources, H.K.; Writing—original draft, K.Y.; Writing—review & editing, C.C.; Supervision, P.K.; Funding acquisition, P.K. All authors have read and agreed to the published version of the manuscript.

**Funding:** This study was supported by research fund from Chosun University, 2022 and This research was supported as a 'Technology Commercialization Collaboration Platform Construction' project of the INNOPOLIS FOUNDATION (Project Number: 1711177250.

**Conflicts of Interest:** The authors declare no conflict of interest.

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
