# Peer review of "A Study on Webtoon Generation Using CLIP and Diffusion Models†"

_electronics, doi:10.3390/electronics12183983_

Round 1

Reviewer 1 Report

Dear authors,

Please consider the following suggestions of improvement:

-A table summarizing and comparing the related work studies could be included in Section 2;

-row 178, please correct the figure number (“Figure 17”);

-row 202, please correct the figure number (“Figure 18”);

-row 247, please correct the figure number (“Figures 19 and 20”);

-Please include the xx and yy axis captions in figures 5 and 6;

-Apart from accuracy and loss, other performance metrics could be presented, as well as the confusion matrixes in Section 4;

-A results comparison between the webtoon genaration developed and other tools could also be presented.

English language use os ok. Some minor issues required.

Author Response

Reviewer 1

comment 1:

A table summarizing and comparing the related work studies could be included in Section 2

answer 1 :

As per the reviewer's suggestion, a performance comparison table for Dalle-2, Stable Diffusion, and GLIDE has been added to Section 2.

comment 2-5 :

- row 178, please correct the figure number (“Figure 17”);

- row 202, please correct the figure number (“Figure 18”)

  • row 247, please correct the figure number (“Figures 19 and 20”);
  • Please include the xx and yy axis captions in figures 5 and 6;

As per the reviewer's suggestion, I have made the necessary corrections to the figure numbers. Additionally, I have added captions to Figures 5 and 6, including details for the x-axis and y-axis. I apologize for any oversight on minor details.

comment 6 :

  • Apart from accuracy and loss, other performance metrics could be presented, as well as the confusion matrixes in Section 4;

comment 7 :

  • A results comparison between the webtoon genaration developed and other tools could also be presented.

I truly appreciate the reviewer's comments. In our previous research, we focused on a GAN-based webtoon generation model, which yielded an inception score of approximately 4. In the current study, our proposed diffusion model achieved an inception score of 7.14. This quantitative improvement in the inception score for webtoons generated in our study indicates higher quality. Furthermore, qualitative assessments through visual inspection also demonstrated superior quality.

However, it's important to note that the data used for experimentation in our study involved continuous sequences of four sentences to make the generated webtoons appear more like actual webtoons. Therefore, direct comparisons with the images generated using the same data were not feasible.

We are currently conducting research on an improved diffusion model to further enhance its performance. Once this research is completed, in accordance with the reviewer's suggestion, we will conduct comparative studies with other models in subsequent research.

I would like to express my sincere gratitude for the valuable comments provided by the reviewer.

Reviewer 2 Report

This study explores the use of deep learning-based text-to-image techniques to enhance creative outputs for webtoon creators. The approach involves converting publicly-available datasets, like MSCOCO, into a multimodal webtoon dataset using CartoonGAN. The study utilizes contrastive language image pre-training (CLIP), which combines muli-lingual BERT and Vision Transformer, to associate text with images. A pre-trained diffusion model is employed to generate webtoons from text and text-similar image inputs. While the technology streamlines webtoon creation, challenges remain in maintaining consistent artistic style and handling muti-sentence and lingual inputs, requiring further research. The work is interesting, and can be accepted with minor revisions.

1. Specify the significance of the research, such as how this approach addresses existing challenges in webtoon creation.

2. Elaborate on the implication of the "inception score of 7.14," explaining what it signifies.

3. How does the use of diffusion models differ from earlies adversarial generative neutral networks in terms of generating realistic images, and what advantages does this approach offer?

4. What is the concept of multimodal learning and its role in enabling AI models to generate output in a format that incorporates diverse modalities? How does this relate to the notion of multimodal artificial intelligence?

5. In what ways does the innovation diffusion theory shed light on the positive impact of AI-driven webtoon creation, and how does this study contribute to advancing the field by utilizing DL-based text-to-image techniques? Pleas provide more details on how the CLIP and diffusion models are employes to generate webtoons form textual descriptions?

 Moderate editing of English language required

Author Response

Reviewer 2

comment 1: Specify the significance of the research, such as how this approach addresses existing challenges in webtoon creation.

answer 1 :

The significance of our research lies in the utilization of text-to-image models in the realm of webtoon creation, aiming to assist webtoon creators in their creative endeavors. While there are numerous tools available that aid in drawing webtoons, such as AI painters, there are relatively few tools that offer support in storytelling and in visualizing the desired form of a webtoon. Many creators, especially those lacking creativity or storytelling skills, could benefit greatly from the text-to-image models investigated in this research when it comes to webtoon creation. I have addressed this in lines 77-85.

comment 2: Elaborate on the implication of the "inception score of 7.14," explaining what it signifies.

answer 2 :

In accordance with the reviewer's feedback, I have incorporated this into lines 322-327.

“The Inception Score is a metric used to evaluate the image quality of generative models. It involves feeding the generated images into a pre-trained Inception model and calculating the probabilities of the images belonging to specific classes[23]. A higher Inception Score indicates better image quality, and the experimental results reveal that the Inception Score for webtoons generated using CLIP and the Diffusion model from continuous text descrip-tions is 7.14. ”

comment 3 : How does the use of diffusion models differ from earlies adversarial generative neutral networks in terms of generating realistic images, and what advantages does this approach offer?

answer 3 :

Generative adversarial networks (GANs) create images through the competitive learning of a generator and a discriminator. The generator is responsible for generating images, while the discriminator assesses whether the generated images are real or fake data. GAN networks update their parameters by minimizing the loss, which quantifies the difference between the generated and real images. GANs also leverage attention mechanisms to produce images that closely resemble real data, and they offer the advantage of faster training. However, GANs can face challenges when one of the networks becomes overly dominant, leading to training instability.

On the other hand, diffusion models involve a process of adding noise to the original data and subsequently recovering the original from the noisy version. Image generation in diffusion models occurs through the process of restoring the original from noise, allowing them to generate data of higher quality closer to real data compared to GANs. The drawback is that diffusion models typically have longer training and generation times than GANs. Despite this, due to their ability to generate high-quality images when compared to GANs, recent generative models primarily favor the use of diffusion models.

comment 4 : What is the concept of multimodal learning and its role in enabling AI models to generate output in a format that incorporates diverse modalities? How does this relate to the notion of multimodal artificial intelligence?

answer 4 :

Multimodal AI refers to the capability of learning and performing tasks involving multiple modalities, which means it can process and generate various types of data such as text, images, and more. In this study, we leveraged two primary modalities: text and images. We extracted distinctive features from each modality and employed the contrastive language image pre-training (CLIP) technique to perform multimodal learning. CLIP facilitates the embedding of both text and image features into a shared latent space, ensuring that similar data points are positioned close to each other while dissimilar ones are farther apart.

CLIP's unique characteristic is its ability to perform zero-shot classification. Given multiple pieces of text and a single image as input, it can classify the image into the appropriate textual category. Conversely, with multiple images and a single text input, it can determine which image corresponds to the provided text. Furthermore, in our research, as demonstrated, we extended this approach by integrating generative models. This enabled us to generate text from images or vice versa, where inputting an image would lead to text generation, and inputting text would result in image generation.

While our study focused on training models for two modalities, the potential for extending this approach to incorporate additional modalities such as sound is a promising avenue. This expansion could enable the AI system to handle an even broader range of tasks and modalities effectively.

comment 5 : In what ways does the innovation diffusion theory shed light on the positive impact of AI-driven webtoon creation, and how does this study contribute to advancing the field by utilizing DL-based text-to-image techniques? Pleas provide more details on how the CLIP and diffusion models are employes to generate webtoons form textual descriptions?

answer 5 :

Innovation diffusion theory, also known as the diffusion model used in this study, is distinct from the innovation diffusion theory proposed by Everett M. Rogers. Rogers' innovation diffusion theory pertains to the adoption and acceptance process of innovative technologies from both individual and organizational perspectives. It explains the stages of acceptance and adoption while considering five key characteristics: relative advantage, compatibility, complexity, trialability, and observability.

- Relative Advantage: This characteristic assesses whether the technology or service offers greater benefits compared to existing ones.

- Compatibility: It evaluates whether the technology's value aligns better with future needs and demands in a specific domain.

- Complexity: Complexity refers to the level of understanding required to grasp the technology.

- Trialability: Trialability indicates the ease with which individuals can experiment with the innovation before fully adopting it.

- Observability: Observability gauges the ability to foresee the outcomes of adopting the technology.

In the paper I referenced, the innovation diffusion theory was applied in the context of implementing artificial intelligence technology within the webtoon industry. The analysis was conducted from the five perspectives mentioned above. The findings suggested that adopting AI-based webtoon generation models (specifically, diffusion model-based generative models) could be viewed positively in terms of innovation adoption by webtoon creators, despite potential issues such as copyright concerns.

Reviewer 3 Report

The paper presents a study on how to employ text-to-image techniques, CLIP and diffusion models to generate webtoon images from textual descriptions. Although is not a high impact breakthrough and there are important issues with the presentation of the results, the proposed methodology as well as the presented dataset, has relevance in the context of the problem of aiding webtoon creators. The major issue with the manuscript is the way it is presented; it needs a complete revision of the content to be suitable for publication. In the following there is a list of major and minor revisions.

1. Overall, a second revision is needed to improve readability.

2. Please do a revision of all the Figures. Most of the images have a wrong numbering, so it is hard to understand what Figure the authors are referring to. Some of them, even if they are identified by context, the description of the Figure in text or the Figure itself doesn’t provide useful information to the reader. Additionally, there are Figures with low quality, such as Figure 1 and 3.

3. It is advised to review Sections 2 and 3.

4. In their experiments, the authors state to use 32 epochs per GPU, but they also state they use 25 epochs. Is this a mistake? Are the authors instead using a batch size of 32?

5. Do the authors used complete MSCOCO? How many images were used for training? What was the split?

6. In Figures 5 and 6, are the authors presenting epochs, steps or what is the x-axis?

7. Tables with results need to be repositioned since they get cropped between pages.

8. There is a lack of analysis of the results, some of them are in the conclusions but seem incomplete. Please improve the analysis of the results.

The authors need to improve the readability of the manuscript.

Author Response

Reviewer 3 :

comment 1:

Overall, a second revision is needed to improve readability.

comment 2 :

Please do a revision of all the Figures. Most of the images have a wrong numbering, so it is hard to understand what Figure the authors are referring to. Some of them, even if they are identified by context, the description of the Figure in text or the Figure itself doesn’t provide useful information to the reader. Additionally, there are Figures with low quality, such as Figure 1 and 3.

comment 3 :

It is advised to review Sections 2 and 3.

answer 1-3 :

I want to express my sincere appreciation for the detailed comments from the reviewer. In accordance with the reviewer's feedback, I have repositioned the images and tables in Sections 2 and 3 to enhance readability. Additionally, I have replaced the images with higher-resolution versions. Your thorough feedback is greatly appreciated.

comment 4 :

In their experiments, the authors state to use 32 epochs per GPU, but they also state they use 25 epochs. Is this a mistake? Are the authors instead using a batch size of 32?

answer 4 : The training was conducted for 25 epochs with a batch size of 128, utilizing four GPUs with 32 batch size per GPU. I made a mistake in my previous writing. I have corrected line 257 accordingly.

comment 5 : Do the authors used complete MSCOCO? How many images were used for training? What was the split?

answer 5 : The data used for training consisted of data from MSCOCO, which was transformed into cartoon-style images using CartoonGAN. After excluding incomplete data, a total of 820,752 text-webtoon data pairs were utilized for training, with 405,040 text-webtoon pairs used as validation data. I have provided details about the data used for training and evaluation in lines 258-261.

comment 6 :

In Figures 5 and 6, are the authors presenting epochs, steps or what is the x-axis?

answer 6 :

To enhance readability, I have integrated Figure 5 and Figure 6. The x-axis represents the total number of training steps. I have also added descriptions to Figure 5's caption for better understanding.

comment 7 :

Tables with results need to be repositioned since they get cropped between pages.

answer 7 :

I appreciate the valuable feedback. To enhance readability, I have repositioned the figures and tables.

comment 8 :

There is a lack of analysis of the results, some of them are in the conclusions but seem incomplete. Please improve the analysis of the results.

answer 8 :

In response to the reviewer's feedback, I have included an explanation of the quantitative metric, Inception score. I am truly grateful for your valuable comments.

Round 2

Reviewer 1 Report

No further suggestions.

English use is ok.

Reviewer 3 Report

Accept in current form

No additional changes